# Students' Technology, Cognitive, and Content Knowledge (TSCCK) Instructional Model Effect on Cognitive Load and Learning Achievement

**Qiong Wu** [1,2]**, Sirirat Petsangsri** [1,]*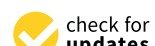 **and John Morris** [1]

[1] School of Industrial Education and Technology, King Mongkut's Institute of Technology Ladkrabang, Bangkok 10520, Thailand
[2] GuiZhou Vocational Technology College of Electronics & Information, Kaili 556000, China
* Correspondence: sirirat.pe@kmitl.ac.th

**Abstract:** The application of scientific and technological approaches in education has been increasing year by year. We evaluated the effect of a TSCCK model based on the cognitive load theory on the cognitive load and learning achievement of vocational students: this model had six components: (1) analysis; (2) content development; (3) cloud development; (4) learning activity development; (5) model implementation; and (6) model revision. We used 62 students randomly selected from 115 students taking an "E-commerce data analysis and processing" course and used cluster random sampling. A total of 31 students were taught with instructions based on the TSCCK model, while 31 students were taught with a traditional method. The instruments used included lesson plans for the TSCCK group developed using the cognitive load theory and the workload profile self-rating scale (WP scale) used to measure the student cognitive load for both groups. The students who learned with TSCCK had significantly lower cognitive load (WP scale) scores than the students who learned with the traditional method, and their achievement scores were higher. The MANOVA confirmed that both the achievement scores and cognitive load measures for the two groups were significantly different at the 0.05 level.

**Keywords:** TSCCK model; cognitive load; learning achievement; cloud



## 1. Introduction

The human world today has undergone constant developments, and the most critical point is the adoption of "scientific and technological means", which greatly advanced human life. The key to the continuous development of science is education, that is, the cultivation of people. For example, in China, because the government and people have attached great importance to education, the country's overall national strength has greatly improved within just a few decades. In education today, a teacher should not only master traditional teaching methods but also integrate "modern information technology" into the classroom and adopt a variety of teaching methods. The "Education Informatization 2.0 Action Plan" issued by the Ministry of Education of China is a new educational concept, "Education Informatization" [1]. This refers to the comprehensive and deep use of modern information technology to promote educational reform and development. Simply put, it requires blending modern new media technology into education to enhance its quality and efficiency. Education informatization has two aspects: teaching informatization and big data. Teaching informatization [1–3] is the application of information technology in all aspects of teaching and enhances its quality and efficiency. Teaching informatization is the core of education information. In teaching process, teachers normally use multimedia, networks, AI, and other technologies to assist their instruction. For example, during history lessons, when teaching ancient Chinese architecture, many magnificent buildings in ancient China no longer exist, sacrificed to the passage of time, but we could reproduce

and experience real historical scenes at that time via multimedia, the internet, AI, and other technologies. By creating teaching scenarios based on multimedia information technology, compared with the traditional teaching mode dominated by instruction, this approach could help students learn better, strengthen their enthusiasm for learning, and create a good atmosphere to learn history. According to psychological and social science studies, this phenomenon refers to cognitive overload or cognitive load. [4–7], which means that "before meaningful learning can begin, the learner may be overwhelmed by a large number of interactive information elements that therefore need to be processed simultaneously" [8].

For example, in lessons about traditional culture in a tourism management major, there is often a single teaching method. Teachers still use a traditional teaching method, namely "infuse teaching", in which they write information on the blackboard for students to learn and memorize. This single, mechanical teaching method is often boring for students [9].

### 1.1. Vocational Education Cloud

In the context of this study, the "cloud" is a resource base, the < 职教云 > or "Vocational Education Cloud" is a collection of applications and databases built by the Chinese Ministry of Education (https://zjy2.icve.com.cn/portal/login.html: accessed on 9 July 2022), to satisfy the requirements of the manual of professional education teaching resource case construction [10]. The cloud is physically distributed over schools, and each teacher has access to it for use in classes. The content could be divided into three parts: before class, during class, and after class. Before class, teachers obtain teaching information resources from the cloud and combine them to satisfy the needs of a particular class. In class, teachers extract relevant resources and functions for "information-based" teaching, including check-ins, brainstorming, playing animations directly from the cloud, etc. After class, teachers assign tests and homework, and the system records student learning trajectories automatically.

### 1.2. Why Is the TSCCK Model Suitable for Vocational Students?

The TSCCK model refers to the following:

- Student technology knowledge (STK);
- Student cognitive knowledge (SCK);
- Student content knowledge (SK).

In recent years, China has witnessed rapid socioeconomic development and an urgent need for many highly skilled personnel. Therefore, specialized training for skilled personnel in vocational colleges has emerged. Higher vocational colleges, as an important part of China's higher education, now emphasize training technical application-oriented talents, distinct from general higher education. For example, teachers in vocational colleges could adopt information technology to organize students in their colleges to learn technical skills as well as general knowledge. In teaching, teachers could effectively design teaching activities and define the major role of higher vocational students [11] by centering on students.

In vocational colleges, the emphasis is on the cultivation of students' operational skills. The TSCCK model focuses on the cultivation of technology knowledge and how to train and improve student technology knowledge. As we all know, vocational school students focus on practical knowledge so that they can directly take up work after graduation. They generally do not need employers to provide pre-job training, thanks to the practical work in school. For example, vocational colleges hire experts in intelligent manufacturing appointed by the cooperative enterprise. They are excellent senior engineers and experts; their experience lets students better understand the current level of professional knowledge, i.e., to introduce cutting-edge technology that the students would need in the future to prepare for practice and employment in the future.

In addition, vocational schools also organize enterprise training for the teacher team from time to time so that they can understand and master current technological frontiers to better connect the teaching content with the industry trends [12]. The cloud for technical support integrates various teaching resources, with teachers as the driver, to design

curriculum teaching activities. The cloud platform and teaching design (TSCCK) model could compensate for difficulties in arranging students' on-site practice, especially for some dangerous and complex concepts that professional workers find difficult to explain on the spot. For example, teaching needs some material support, and these materials are often expensive. Vocational college funds are limited, and sometimes, the schools cannot afford to construct a mechanics training room. Without a training room, teachers could use information technology to simulate the real environment in class while the cloud provides technical support. For example, by teaching an "architectural mechanics" course at the initial stage, teachers could use the cloud to display pictures and videos of a construction site to create a more realistic and safe training environment [12]. In addition, during teaching, the three basic ideas of the TSCCK model: technology (STK), cognitive (SCK), and content knowledge (SK) could be combined and integrated by teachers. In this way, professional teaching and industry are closely interconnected, so the teaching effect of vocational college courses is fundamentally improved.

Based on the cognitive load theory in vocabulary acquisition, our study developed a TSCCK model using the cloud; a key aim was to decrease cognitive load and enhance learning. As examples of the TSCCK model instructional steps, we describe the design fundamentals of an "E-commerce data analysis and processing" course, where we improved the course activities using the cloud as a primary resource. Given the improvement in learning we achieved, we consider that it lays a foundation for further research on this model in other areas of technology-based teaching.

### 1.3. Previous Work and Background

1.3.1. Cognitive Load Theory in Instruction

The cognitive load theory [13] uses modern cognitive psychology research to show how to facilitate learning and teaching design; it involves a wide range of application capabilities and operational values and leads to positive development. The 21st century is an era of information. The times require people to "learn how to learn"; to absorb key knowledge from a vast ocean of information, one must learn quickly and efficiently. This is a basic characteristic of modern learning, as well as an essential quality and pursuit of modern learners. The main purpose of this study was to investigate how to promote the germane load of students under the background of modern information-based teaching within the TSCCK model.

It is required not only to adopt an appropriate teaching design to support students and lower both extraneous and intrinsic cognitive load [13] but also to enable students to learn how to use germane cognitive load. For this reason, in the context of modern information technology, teachers need to read research on cognitive load to strengthen the management of student cognitive load and efficient teaching.

First of all, in teaching, the manner of speaking of teachers directly influences students' cognitive processing and load when listening to lectures. For example, when teaching mathematics, many concepts are abstract. While listening to lectures, students consume cognitive resources and feel tired if they use abstract thinking for cognitive processing. Thus, mathematics teachers should express themselves in a clear, accurate, concise, and logical manner when designing classroom presentations [14] to ease the burden of language cognitive processing.

Secondly, to reduce cognitive load, sample teaching [15] was found to be a good teaching model, which could lower the cognitive load and enable students to pay more attention to the general structural characteristics of problems, as well as the principles, rules, and algorithms to be adopted in specific circumstances, so as to enhance students' schematic and automatic acquisitions [13]. For example, when teaching statistics in the sixth grade of elementary school, teachers may select real-life examples that the students are very familiar with [14]; for example, their daily schedule structure or the family's living expenses plan, or other projects based on simple demographic details. These examples were closely associated with student life and study, as well as the schema stored in their long-term

memory and relevant knowledge and experience. The students were familiar with them, so their cognitive learning load was reduced. Therefore, during teaching, examples familiar to students were used as subjects, which improved the ease of learning significantly.

To this point, the courseware design and application of the cognitive load theory need to be addressed. According to the theory, extraneous cognitive load is associated with the organization and presentation of materials. Thus, in order to reduce the load intensity, when designing courseware, one should organize relevant teaching materials reasonably. For example, when teaching quadrilaterals in eighth-grade mathematics [14], teachers were advised to design visualizations showing the correlation and differences between various quadrilaterals, e.g., general quadrilaterals, trapezoids, parallelograms, rectangles, rhombuses, and squares; a comprehensive diagram could describe the logical relationship between the various quadrilaterals visually and intuitively, and teachers could use the diagrams and text together during the lesson. In this way, not only could the resources of student cognitive processing be saved and their inherent cognitive load lowered, but students could also build a correlation between the speech and image models so as to facilitate their meaningful learning.

### 1.3.2. TSCCK Background

Considering the cognitive load theory (CLT) is very important when designing teaching programs. In particular, we should consider the participation and identification of users when adopting technological means for cognition [8]. Learners' working memory ability could influence their cognitive load [16]. The cognitive load theory considers that adopting different teaching designs and learning materials may lead to different cognitive loads and experience styles and influence learner interaction styles because the individual working memory capacity is limited [16,17]. Therefore, to improve learning, it is necessary to explore the difficulty in learning materials and teaching design. Images are not only an important intermediary between languages and concepts but also a mode of physical experience [18]. The working memory and long-term memory of learners are part of the cognitive load theory. This theory holds that since working memory has a limited capacity, the cognitive structure of learners' working memory could be integrated through reasonable teaching design to lower the cognitive load [16]. The way learners interact and experience the learning environment is influenced by teaching design and the difficulty in learning materials [16,17]. The cognitive load theory is considered one of the most significant theories in teaching design. When applying technology to teaching design, teachers should identify the cognitive processes that learners activate [16]. Mayer [19,20] introduced the principle of spatial continuity, which showed that if learners used both text and photographs in the learning of words, this would lower their cognitive load and improve their learning effect, compared with those who used text alone in a multimedia learning environment.

Accordingly, compared with learners with higher cognitive loads, learners with lower intrinsic cognitive loads tend to occupy less working memory capacities, and information could be converted into long-term memory for storage [21]. Galy & Melan [22] posited that the total quantity of resources in three categories was cumulative and fixed. If the extraneous cognitive load includes more resources, there would be fewer resources in the working memory, and it would be hard for learners to convert short-term memory into long-term memory as they learn. By examining the application of the cognitive load theory by some instructional designers to structured design fields (generally well-structured designs and complex ill-conditioned designs), we could see that researchers would be able to replicate more realistic research into a problem-solving context and identify the domains that are more applicable to a variety of strategies. Further, instructional designers in charge of training would have a better understanding of embedding the cognitive load theory in situational learning [23]. In an effort to respond the problem of the cognitive load effects in fields other than mathematics and science, empirical research is required to uphold the role of cognitive load [24,25].

For example, Oksa et al. [26] discovered that when high school freshmen were provided with modern English annotations while learning Shakespeare's plays, their cognitive load performance was lowered. Si et al. [27] observed that undergraduates, who solved programming problems with worked examples, managed cognitive load better and thus better constructed and automated schemas. Several other cognitive load effects were observed in other studies, including the minimization of distraction through the simultaneous presentation of materials [28], expertise reversal effect [26,29], and benefits of fading steps within a solution as learners acquired problem-solving skills [27].

## 2. Methods

Our study used the following steps:
- Identify the research objectives;
- Clarify the details with the population and the experimental class;
- Describe the research instruments: including the lesson plans, WP scale, and e-commerce data analysis and processing test.

### 2.1. Research Objectives

We compared the cognitive load and learning achievement between students learning with the TSCCK model and those who learnt with the traditional method.

The hypotheses were:

Students who learnt with the TSCCK model

H1a: had lower cognitive load and

H1b: higher learning achievement.

### 2.2. Population and Sample

The details of the population and the experimental class are set out in Table 1.

**Table 1.** Details of the population and the experimental class.

| | | |
|---|---|---|
| **School Location** | Kaili | 107.99848° W, 26.58727° N |
| **Name** | GuiZhou Vocational Technology College of Electronics & Information | |
| **Course** | E-commerce data analysis and processing | |
| **Duration** | 4 weeks | Second semester of the 2021–2022 academic year |
| **Student age range** | 18–21 years | |
| **Level** | Second year | |
| **Total population** | 115 | |
| **Sampling strategy** | Select 62 students from 115 students using cluster random sampling | |
| **Class size** | Experimental (with TSCCK) | 31 (4 female and 27 male) |
| | Traditional | 31 (9 female and 22 male) |

There were two variables in the study:

1. The independent variable: the TSCCK model with the cloud versus a traditional method.
2. The dependent variables: students' cognitive load and learning achievement.

The "E-commerce data analysis and processing" course combined theory and practice; it was designed to teach the basic concepts of e-commerce data analysis and processing systematically. Students became familiar with the analysis of online shop data, including data collection, data sorting, data analysis, data processing, data visualization, etc. This course was suitable for students majoring in e-commerce.

### 2.3. Research Instruments

2.3.1. Lesson Plans

A major part of the preparation for this research was the development of lesson plans specifically designed to reduce student cognitive load. The materials for the formal lectures followed the college curriculum, and the teaching and assessment methods were the same as those for the formal lectures. The lesson plans designed based on the TSCCK model are in Figure 1 and described in detail as follows.

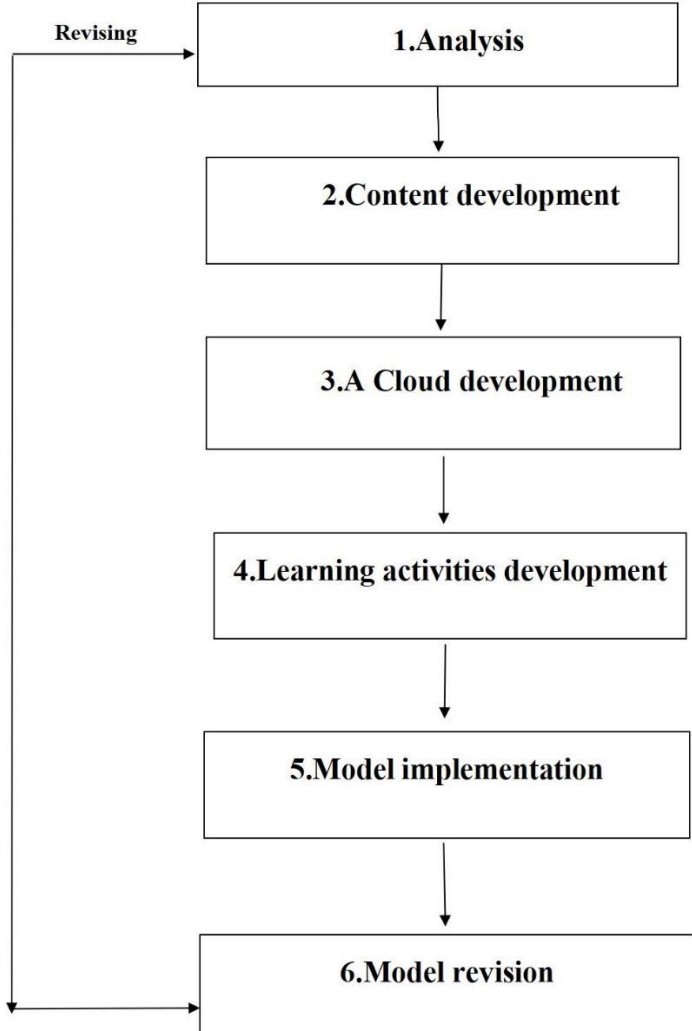

**Figure 1.** Development of the lesson plans following the TSCCK model.

1.    Analysis

(1)    Analyzing the objectives.

The teaching objective was the outcome or standard expected by the teachers and students. To allow the TSCCK model to play a better role, we first analyzed the teaching objectives and examined the existing curriculum.

(2)    Analyzing the content.

As a key part of the TSCCK model, the teaching content refers to the material that teachers explain to students in the classroom, and teachers should identify the difficulties in the content based on the real student capabilities. For example, in our research, we identified "customer profile analysis". We first tested students' relevant knowledge to determine the key points and difficulties of the class. Before discussing e-commerce data

processing, we first selected simple data processing cases and provided them to the students to test through the cloud, and then analyzed the results. It was found that the functions in Excel (e.g., IF (), SUMIF (), RANK (), etc.) revealed the students' biggest shortcoming, allowing us to establish the most important and challenging teaching strategies.

(3) Analyzing the learners.

The TSCCK model is student-centered. Thus, teachers should fully understand the student's abilities before setting the course because the level of student abilities could help teachers set the starting level of the course and adopt the appropriate teaching strategies to offer support in this course.

(4) Analyzing the context.

Using the cognitive load theory, we applied the TSCCK model to the cloud. Here, it was vital that the cloud be used to reduce the cognitive load effectively. In this case, by releasing cases on the cloud, we let students think about how we could achieve precision in product promotion, that is, promote products mainly to people in need. At this time, it was necessary to analyze the consumer group demographics, i.e., age, gender, income level, education level, region, etc.

2. Content Development

(1) Design Fundamentals.

- Analyzing the resources.

The learning resources required to satisfy the objectives of this course needed to be identified. These included the theoretical knowledge and practical skills needed for e-commerce data analysis. As an example, we chose the "1 + X E-commerce Data Analysis Vocational Skill Level Certificate" of the Beijing Bo Guiding Future Company; from the cloud (see Section 1.1), we could find a page listing the courses in the E-commerce department of our university, followed by further menus pointing to resources for these courses (Figure 2), including the "TaoBao Shop" backend shown in Figure 3.

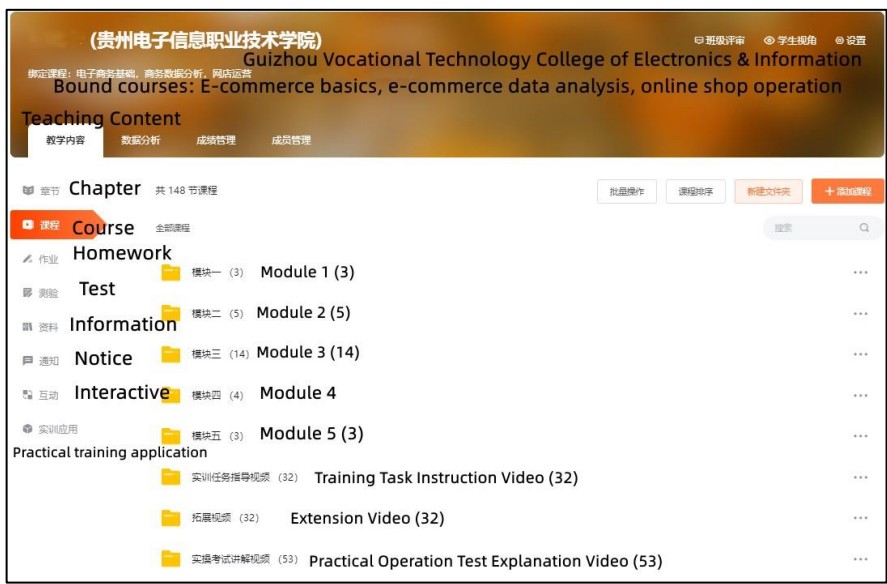

**Figure 2.** Theory and practical training guide video collection.

- Defining the instruction characteristics

The styles of instruction that help to reduce cognitive load [30] are:

- Worked examples and problem-solving;

Here, the cloud enabled a large number of diverse worked examples to be presented rapidly and effectively and allowed the rapid visualization of the results. This reduced the cognitive load and promoted learning.

- The reduction of extraneous load and the management of intrinsic cognitive load;

Extraneous load is related to the teaching design and needs to be managed by the teacher. Intrinsic load is added if examples are too complex and involved. Visualizations are a significant way for teachers to lead students, step by step, through complex problems. Through cloud technical support and appropriate teaching methods, the presentation of learning materials changed and lowered the intrinsic and extrinsic cognitive load. For example, we taught advertising delivery using "Taobao" (a commercial e-commerce website, https://world.taobao.com/, which provides a back end that we were able to access). Using the backend of the "Taobao Shop", students could search the rankings for goods; an example of the output is provided in Figure 3, which includes considerable real marketing data. By looking at the detailed data, students were able to obtain a better "picture" of the needs. This rendered teaching more intuitive and simple because it reduced both the intrinsic and extraneous cognitive load.

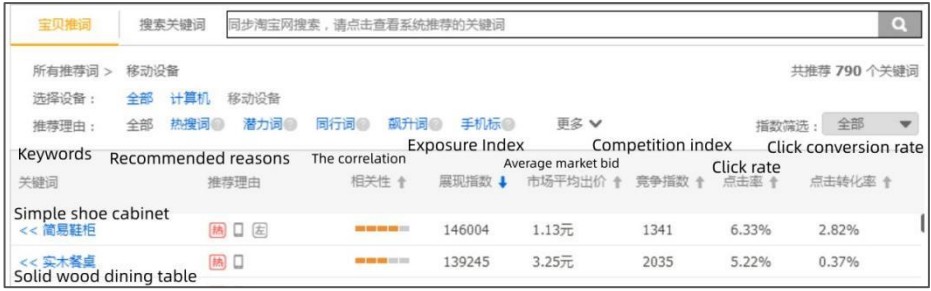

**Figure 3.** Ranking of the various indicators in "Taobao" to teach product promotion.

- Determining the fundamentals.

To achieve the desired teaching effect, the teachers needed to evaluate the "E-commerce data analysis and processing" teaching by using the test cloud. For example, in order to understand how students mastered the skills of the "paid promotion time-region analysis", we imported questions into the testing system, and students directly answered the questions online. By analyzing the results, we are able to see if students mastered the necessary skills (Figure 4).

(2) Instructional fundamentals

To script an example lesson in the "Paid Promotional Time-Geographic Analysis" course, we split the instructional fundamentals into five sections:

- Task;

Using the cloud, the students were presented with learning tasks, which were simple and easy to follow to reduce the cognitive load. For example, many spreadsheet operations were needed in this task. Here, we released learning videos of some simple spreadsheet operations (such as the average, sum, etc.) through the cloud so that students could learn by themselves first, supported by feedback from the "cloud" system, further reducing the cognitive load.

- Process;

In this stage, to enable the students to learn practical concepts and finish the tasks successfully, we emphasized key points and mistakes. For example, from students' feedback before class, the problems and mistakes encountered by students in the pre-class learning were recorded; see a typical report in Figure 5, which counts the number of students that found the MID, LEFT, etc. functions difficult to understand. Then, the MID and HOUR functions, which were difficult in this class, were emphasized.

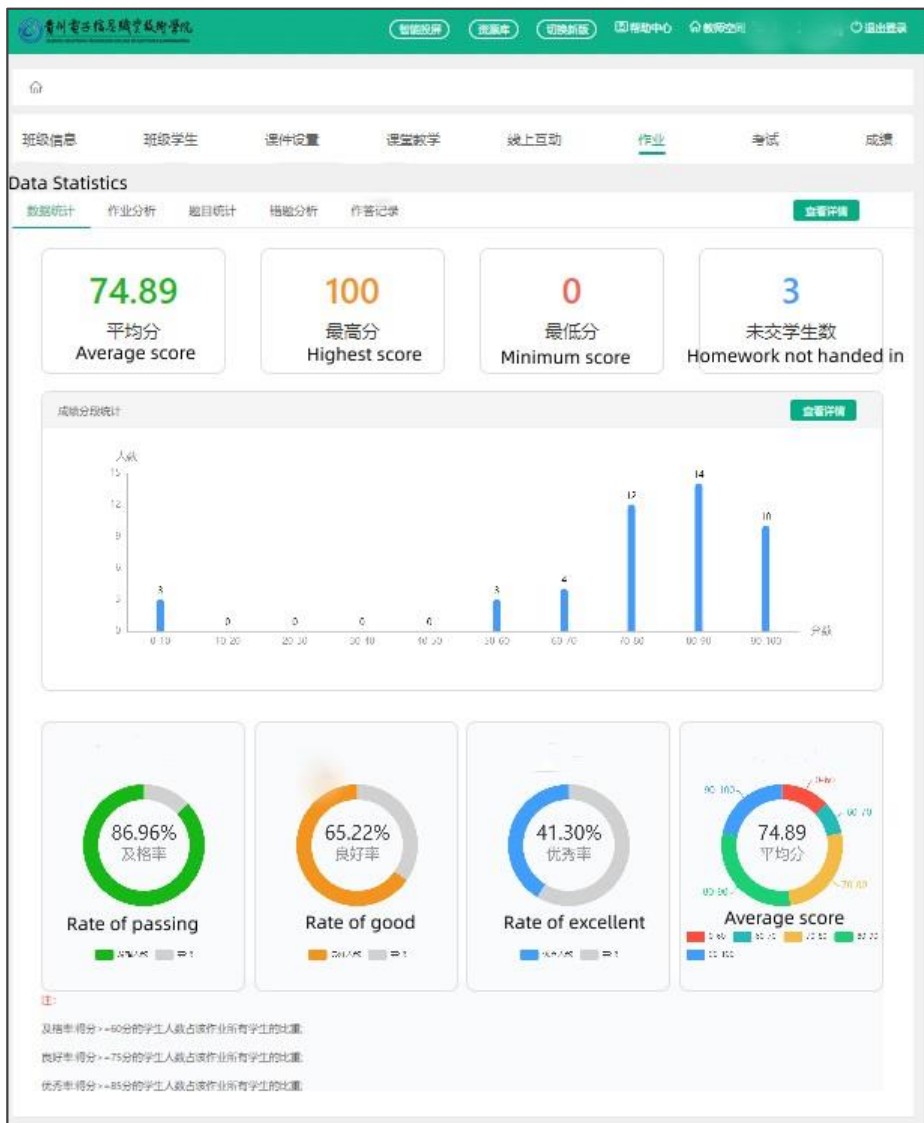

**Figure 4.** The cloud test system shown graphically with various student performance indicators.

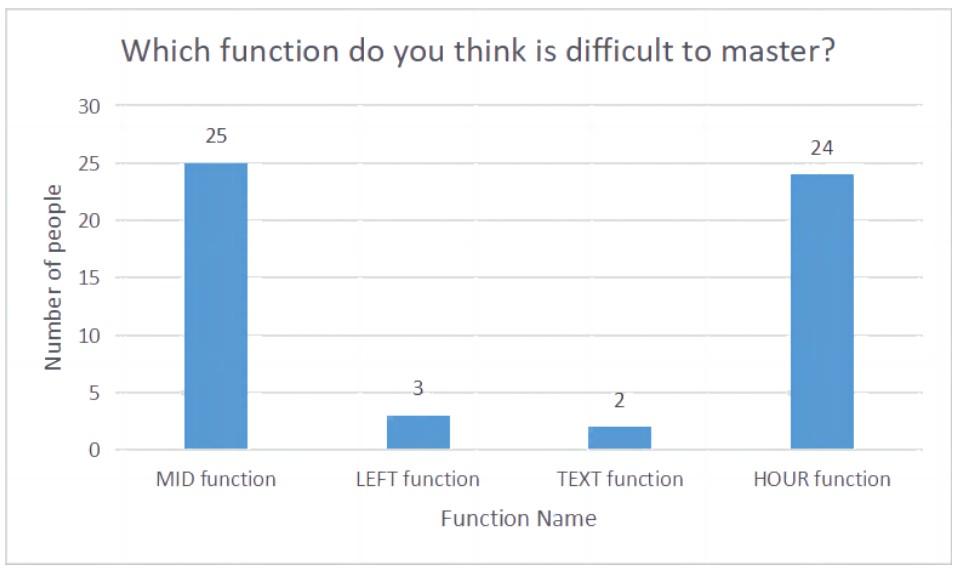

**Figure 5.** Counts for the key mistakes recorded by the cloud.

- Resources;

In this stage, we provided the materials required to complete the learning tasks. In terms of resource learning, the content included theory and practice, learning videos, a question database, and the practical operation of the system. The system also allowed students to find homework exercises and other practical tasks, insert answers to the questions (which are then checked by teachers), and perform some additional functions.

- Evaluation;

We showed students the tests offered by the cloud and explained to students how the cloud evaluated them, which included attendance, classroom questions, interactions, and classroom practice. The evaluation records were stored in the cloud for later automatic calculations after the students finished a class.

- Conclusion;

In the fifth step, the course design also included the students' own practice; that is, we provided them with an opportunity to explore what they learnt by themselves. For example: when the lecture was finished, the students in their "Taobao" shop were encouraged to continue the practice, analyze the real store data, find the customer demographics, understand the store location, time, etc., and from the data, determine an optimum time and place of advertising. In the end, the student results were kept in the cloud so that the teacher could review them.

- Revising the process.

At this point, we revised the teaching design because teaching is student-oriented, and students are individuals with different characteristics, so the teaching design should be modifiable at any time.

3.　Cloud Development

The cloud, described in Section 1.1, is a resource portal for Chinese teachers and students. A brief description of its use follows.

- Opening a new class

A teacher follows the menus to enter a new course and its details, including the classroom instruction, instruction design, course content, and requirements. The class lists and other administration details are also added.

- Teaching process

    - Before class, the students learnt independently;

Before teaching, the cloud simplified learning tasks and lowered cognitive load. Teachers allowed the students to log into the cloud in advance and asked them to complete the tasks assigned through the rational teaching design by releasing tasks to them. In doing so, the students were inspired, and their cognitive load was also reduced. For example, when talking about "customer service data analysis", relevant learning videos would be sent to the cloud so that students could learn independently first [31].

    - In-class activity involved the implementation of student task;

In view of the uniqueness and difficulty of the e-commerce data analysis and processing course, the teachers introduced simple knowledge at the beginning and then gradually raised the difficulty. For example, through pre-class learning, the tasks continued to be published on the cloud, including the task materials, background, requirements, and steps. Furthermore, students were asked to use functions to calculate the consulting conversion rate and the equal score of each customer service staff according to the task requirements. Using an exploratory learning style allows the students to practice first, and then the teacher explains.

    - After class, the students designed and created.

Teachers should keep measuring the professional knowledge of learners according to their performance and cognitive load and adjust their teaching dynamically to the needs

of individual learners. For example, if it is found that some students are too slow to learn the practical operations in the course of the class, they could explain this part to the students separately in advance and adjust the teaching methods. By using the cloud, the students were able to integrate three kinds of knowledge: student content knowledge (SK), student cognitive knowledge (SCK), and student technology knowledge (STK). The students established a TSCCK framework by finishing the related learning tasks.

4. Learning Activity Development

(1) Designing the instructional materials

- Designing the learning activities;

The TSCCK model emphasizes that students are the center of the class. During the teaching of "E-commerce data analysis and processing", the ability to attract students' attention was a problem to be pondered by teachers. The unique goal of the TSCCK model is to reduce cognitive load and improve learning by using the cloud.

- Determining the instructional steps;

To allow the teaching of the "Data analysis and processing" course, supported by the cloud, to proceed smoothly, teachers designed the steps of the teaching activities carefully, not only reducing the students' cognitive load but also stimulating their learning interest.

- Designing the assessment.

Teachers could use the technology from the cloud to evaluate students' learning progress. The cloud would record, for example, the completion of each student's automatic homework, check-in status, whether they arrived late or left early, and answer questions quickly. These could be adopted as grades and formative assessments. Students' mid-term test scores and final scores could also be automatically displayed on the cloud for summative judgments.

(2) Developing the instructional materials

- Using the electronic classroom;

When the teacher imparted their practical skills to students, they would use the functions of the electronic classroom and control the screen when explaining it. The students could see the teacher's actions directly by looking at the monitor in front of them [32]. After the teacher finished the explanation and clicked the release function, the students could operate by themselves.

- Developing "E-commerce data analysis and processing" test and WP scale;

The assessment tools were a multiple choice test and a WP scale, i.e., a cognitive load test.

- Developing the lesson plans.

The instructional steps for the TSCCK model were:
Step 1: Stimulate students' prior knowledge;
Step 2: Guide the students to follow the teacher;
Step 3: Deal with the operational tasks;
Step 4: Teach new tasks;
Step 5: Assign operational task tests.

5. Model Implementation

In this step, the teachers put their instructions on the e-commerce data analysis and processing course into practice, which involved the preparation of instructions for students, including the classroom setting and learning materials. The suggested instructional steps in the classroom were as follows:

(1) Stimulate students' prior knowledge;

As a teacher, one should inform students of two points before teaching. The first is the simulation of previous knowledge, and the second is the learning content, objectives, and effect in the class.

(2)     Guide the students to follow the teacher;

The teacher should introduce the learning tasks to the students, as well as the precautions for the operation process. The students should carefully listen to the instructions and complete the set tasks.

(3)     Deal with operational tasks;

During learning, the teacher plays the role of providing a scaffold and helping students at critical moments. The students work in groups and finish the learning tasks via group learning.

(4)     Teaching operational tasks;

During learning, the teacher plays the role of the "troubleshooter" and urges students to perform the correct operations. The students discuss the methods and operation steps to finish the learning tasks in groups.

(5)     Assign operational tasks.

As the last step, the teacher assigns the operational tasks and student exercises and provides individual tutoring.

6.     Model Revision

Teachers should study the WP scale measurements to determine the effectiveness of the model. Revision is a constant process. Whenever teachers find parts that are hard or unclear for learners, the lessons should be revised and adjusted to help learners better achieve the instructional goals.

### 2.3.2. Cognitive Load Tools (WP Scale)

Tsang & Velazquez [33] described a WP self-rating scale for assessing cognitive load. The scale had seven dimensions, set out in Table 2. The methods we used are further described in Section 4.

**Table 2.** Dimensions of the WP Scale.

|   | Resource | Meaning of the Dimension |
|---|---|---|
| 1 | Central processing | Mental resources for the task selection and execution |
| 2 | Response | Mental resources for responding to the task |
| 3 | Spatial coding | Mental resources for the spatial activity of the brain during the task |
| 4 | Language coding | Mental resources for the speech activity of the brain when completing a task |
| 5 | Visual reception | Mental resources to obtain information from visual channels to complete a task |
| 6 | Auditory reception | Mental resources to obtain information from auditory channels to complete a task |
| 7 | Operational | Impact of moving the limbs on cognitive load to complete a task |

Their WP scale required learners to evaluate the psychological resources occupied by completing a task. They assigned a number between 0 and 10 for each of the seven psychological resources based on their subjective feelings after completing the learning task. "0" implied that the resources were no occupancy at all, and "10" indicated that the resources were of maximal occupancy [34].

### 2.3.3. "E-Commerce Data Analysis and Processing" Test

We used a multiple-choice test to assess learning achievement. The test consisted of 25 multiple-choice questions with a full score of 100 and was evaluated by experts in the relevant field before the formal test; the indices of congruence (IOC) and reliabilities (KR-20) were determined from their responses, and the reliability values were analyzed with SPSS as shown in Table 3.

**Table 3.** Consistency and reliability of the e-commerce test and WP scale.

|  | Index of Congruence | Reliability (KR-20) |
|---|---|---|
| E-commerce Test | 1.0 | 0.957 |
| WP scale | 1.0 | 0.898 |

*2.4. Data Collection*

The WP scale test and the "E-commerce data analysis and processing" test were used to assess each student's learning cognitive load and learning achievement after the class for both groups after week four. A MANOVA test assessed the relationships between students' learning achievement and cognitive load of an experiment group and a control group.

**3. Results**

The data of the experimental group and the control group were compared with various tests. For all tests, the significance level was set at 0.05.

Table 4 shows the data for the experimental and control groups using various tests. First, we computed the means and standard deviations, followed by the Shapiro-Wilk test, which indicated that the distributions were highly skewed. Therefore we computed and used the medians, indicated by *M*, as the main basis for the comparison. For learning achievement, the TSCCK group (mean = 55, SD = 25, *M* = 52.0) had a better (i.e., higher) score than the control group (mean = 10, SD = 16, *M* = 4). For cognitive load, the TSCCK group (mean = 32, *M* = 31.0) also had better (lower) WP scores than the control group (mean = 53, *M* = 53). The difference between the TSCCK group and the control group was large. We note that several factors in the TSCCK approach likely combined to produce significant benefits. However, the cognitive load was clearly significantly lower, so we attribute most of the benefit to lowering it. Although overall, as seen via the improvements in the median score, the improved score was not uniform; some students benefited more than others. This is further discussed in the conclusion section. Several reports [35,36] suggested that for sample sizes >30, it is reasonable to assume normality and use a MANOVA in the next section to confirm that the distributions of our samples were, in fact, significantly different.

**Table 4.** Post-test results: learning achievement and cognitive load after course completion.

| Variables | | N | $\bar{x}$ | SD | Med | Shapiro-Wilk | | F | Sig | Levenne Test |
|---|---|---|---|---|---|---|---|---|---|---|
| Dep | Ind | | | | | W | p | | | |
| Learning achievement | TSCCK | 31 | 55 | 25 | 52.0 | 0.909 | 0.012 | 67.684 | <0.001 | <0.001 |
| | Control | 31 | 10 | 16 | 4.0 | 0.657 | <0.001 | | | |
| Cognitive load | TSCCK | 31 | 32 | 5 | 31.0 | 0.924 | 0.030 | 393.80 | <0.001 | 0.025 |
| | Control | 31 | 53 | 3 | 53.0 | 0.959 | 0.270 | | | |

$\bar{x}$, mean; SD: standard deviation; Med: median; W: Shapiro-Wilk W; p: Normality probability; F: F distribution; Sig: significance (from the MANOVA).

To confirm that the differences were statistically valid, we used a MANOVA to test the two dependent variables, i.e., learning achievement and cognitive load. We started with Bartlett's test of sphericity, which reported <0.001 (Table 5), indicating that learning achievement was strongly correlated with cognitive load. For further validation, we used a Box M test, which also showed no significance (Sig > $\alpha$).

**Table 5.** Testing the dependent variables.

|  | Box's M Test | Bartlett's Test |
|---|---|---|
| Learning achievement and cognitive load | 0.068 | <0.001 |

As shown in Table 6, the MANOVA used Wilks' lambda, Hotelling's trace, Pillai's trace, or Roy's largest root, which were later converted to F statistics to assess the significance of the differences between the TSCCK group and the control. In Table 5, the significance level (Sig > α) was set as a = 0.05. Also, Bartlett's test of sphericity was <0.0005, or statistically significant, confirming a high correlation and the suitability of the prediction [35]. In Table 6, Wilks' lambda had sig < 0.001, or much less than our $p < 0.05$, again confirming that the TSCCK model had at least one variable that differed from the traditional group. The MANOVA results appear in the final columns of Table 4. These tests confirmed a statistically significant difference in the learning achievement and cognitive load between the groups.

**Table 6.** MANOVA test results classified by group.

| Variable | Statistic | Value | F | Sig. |
|---|---|---|---|---|
| | Pillai's trace | 0.902 | 270.406 [b] | <0.001 |
| | Wilks' lambda | 0.098 | 270.406 [b] | <0.001 |
| Group | Hotelling's trace | 9.166 | 270.406 [b] | <0.001 |
| | Roy's largest root | 9.166 | 270.406 [b] | <0.001 |

[b] Exact statistic.

## 4. Discussion

Our results showed that there were significant differences at the 0.05 level in learning achievement and cognitive load between the TSCCK and control groups (Table 4). The support from the cloud was a significant factor in this result.

With its help, if teachers adopt appropriate teaching methods and knowledge of the three aspects of the TSCCK (the cloud, the TSCCK instruction model, and the cognitive load theory), they could be developed appropriately, especially using the cognitive load theory.

In similar previous work, Zhang [37] used a cloud network to design courses, homework, and relevant teaching materials to construct online courses; Zhang believed that the sharing of cloud network teaching resources broadened the vision of teachers and students and encouraged high-quality teaching. Yin et al. [38] also put forward a three-stage smart classroom teaching process before, during, and after class, believing that it effectively facilitated the reform and innovation of educational concepts and modes, teaching contents and methods, and accelerated talent cultivation. Zhang [39] held that the vocational education cloud rectified the defects in the traditional teaching model. Chen et al. [40] claimed that an online hybrid teaching scheme based on the vocational education cloud provided full play to the functions of teaching, realized a deep fusion between information technology and the curriculum, promoted students' deep learning, facilitated students' grasp of theory and practical skills, improved the learning state and experience, and trained autonomous and life-long learning abilities. Pan & Tao [41] described an advanced information-based teaching management cloud to introduce teaching management in modern apprenticeship classes, exploring and resolving problems related to modern apprenticeships in classroom teaching, for example, matching business owners with school teachers, dividing specific teaching tasks in the class, and providing play to inherent gifts, to achieve the best teaching effect. They considered that the cloud-based classroom correctly guided students to approach several problems in electronic circuit design, which effectively cultivated innovative and practical abilities. They included diversified forms of classroom interactions, such as mobile phone check-ins, discussions, and quick responses, so that students' interest in learning was stimulated and the class efficiency was improved. Gao [10] argued that a classroom' constructed from a "vocational education Cloud" + "Tencent Classroom" would ensure that online learning and offline classroom teaching would have equivalent quality and promote innovation in a teaching model to achieve new results that were significant. Zhuo [42] believed that with information techniques, for example, the cloud

classroom, teaching reform would be advanced, thereby enhancing the efficiency and quality of teaching and cultivating more high-technology and skilled personnel.

## 5. Conclusions

To summarize, the TSCCK model based on the cognitive load theory aims to promote the cognitive load and learning achievement of vocational students. This model has six key steps: (1) analysis, (2) content development, (3) cloud development, (4) learning activity development, (5) model implementation, and (6) model revision. Our results confirmed our hypotheses: the TSCCK model led to lower cognitive load (H1a) and higher learning achievement (H1b). This was demonstrated with the assessments of lower cognitive load in students coupled with higher scores in an e-commerce data analysis and processing test.

Our study leads to recommendations based on (a) the pedagogical implications of the instructional model and (b) further research.

### 5.1. Pedagogical Implications

1.  In planning, teachers should consider students' cognitive load and associate it with their prior knowledge, which is essential for increasing their learning enthusiasm and attention to classroom activities;
2.  Therefore, to achieve the benefit of the cloud with the TSCCK model, teachers should conduct a preliminary study to examine the differences between students' theoretical and practical knowledge for applying the technology;
3.  Most of the learning content is stored in the cloud. During teaching, especially when students perform tasks independently, teachers should understand the students' grasp of knowledge from that task, from the results shown on the cloud, and adjust their strategies appropriately.

### 5.2. Recommendations for Further Research

1.  There were several factors in the TSCCK model, and this study found that the following combination was effective: a factor analysis to determine the relative importance of each factor may elucidate areas to focus further improvements;
2.  In particular, the strong benefit observed in our study may be specific to our course, "E-commerce data analysis and processing", and further work to identify factors for other courses is needed;
3.  The present study discussed the cognitive load and learning achievement of vocational students. Identifying specific operational skills should be studied to guide the extension of the model to other disciplines.

**Author Contributions:** Conceptualization, Q.W.; methodology, S.P.; validation, J.M.; formal analysis, J.M.; investigation, Q.W.; resources, Q.W.; writing—original draft preparation, Q.W.; writing—review and editing, J.M.; supervision, S.P.; project administration, S.P.; All authors have read and agreed to the published version of the manuscript.

**Funding:** This research received no external funding.

**Institutional Review Board Statement:** Not applicable.

**Informed Consent Statement:** Informed consent was obtained from all subjects in the study.

**Data Availability Statement:** Our data is available upon request from the corresponding author.

**Acknowledgments:** We thank Shoufeng Wu from Guizhou Vocational Technology College of Electronics & Information, China, for their comments on this paper.

**Conflicts of Interest:** We declare no conflict of interest.

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
