# Peer review of "Students’ Technology, Cognitive, and Content Knowledge (TSCCK) Instructional Model Effect on Cognitive Load and Learning Achievement"

_education, doi:10.3390/educsci12120916_

Round 1
Reviewer 1 Report
In this paper the authors present a comparation of results from 31 vocational students taught based on TSCCK model versus 31 students taught with the traditional method, in the context of an existing cloud with digital resources.
The text is very long, with several repetitions, but the description of the comparation study is incomplete and should be improved.
For instance, the sample method is not clear, if there were 115 students (which seems like the Universe of the study) why only 62 were involved in the study? which were the sample criteria involved?
This text doesn't discuss the results and not conclude if the hypothesis was rejected or not. The hypothesis should be expressly formulated.
The text organization uses too many subsections in 4 levels which seems excessive. Some of them have only one or two paragraphs (see page 11).
The title is too long and complex, please simplify.
I cannot understand the justification for using parametric texts. In page 12 we can read "the center limit theorem said that if the sample size is big(n>30). the data distribution was almost normality", why can't you just text for normality instead of considering that a sample of 30 should be enough?
The results show huge differences between the instruments results for the two groups. Don't you have any other explanation for this big difference than the difference in teaching methods? The traditional methods seem too bad to be truth.
In the conclusions the authors mention that "the research instrument used is tested less frequently", it is not clear to me what this means.
Typos to correct:
pag.1 "The samples were selected 62 students" check the English language
pag.1 ", 6) model revision and students who learned with" some punctuation mark seems to be missing
pag.4 "Galy & Melan [23] posited that the total quantity of resources is fixed" please review
table 1 in the title of the columns the 3rd has geographic coordinates "107.998482 W,26.587275 N", please explain the meaning
pag.12 " Below are the test results" there is no table below this line
pag.13 "This indicates a 5% risk of concluding that an association exists between learning achievement and cognitive load when there is actual association. " this is not an error and has no risk.
pag. 13 "which as per the preliminary agreement" please revise
Reviewer 2 Report
Dear authors,
Thank you very much for your manuscript.
After reading your manuscript, I would like to suggest the following improvements:
1. Page 2, paragraph 6. Why there is a question mark at the sentence “The TSCCK model focusses on the cultivation of technology knowledge and how to train and improve student technology knowledge?”
2. Double “sample teaching” at page 3 paragraph 6.
3. The school information in Table 1 should be concealed with an appropriate label, as it may raise ethical concerns in research.
4. To aid the reader's comprehension, 2.3.2.2 should also be supplemented with figures and illustrations.
5. Please justify the use of Shapiro-Wilk as opposed to Kolmogorov-Smirnov for the normality test.
6. Please revise Table 4, as the data do not make sense for the elaboration of normality. Which information pertains to the experimental group and which to the control group?
7. Please conduct Levene's Test to determine and evaluate the comparability of the experimental and control groups.
8. Discussion is too superficial. Please elaborate on this study's findings and contribution to the body of knowledge.
9. A professional editor is required to assist the authors in simplifying a portion of the manuscript so that it can be understood by the majority of the scientific community.
Thank you. I hope the authors will find the feedback helpful in improving the quality of their manuscript.
Reviewer 3 Report
The research seems up-to-date and interesting in terms of its subject. Although the research title is too long, it is suitable for the purpose. But the title should be shortened and checked for grammar.
It was also good that the abstract of this research was in the form of a structured summary. In other words, it would be good to write a summary that includes the purpose, method, data collection tool and summary of the results. However, the research findings should be briefly mentioned in the abstract, and it would be more appropriate to give the findings without numbering.
The introduction of the research is appropriate in terms of literature.
The introduction part of the research is sufficient in terms of subject area. The bibliographies used are up-to-date. Therefore, the use of new bibliography in the introduction and discussion sections of the research has enriched the research.
Purpose and sub-objectives should be written before the method in accordance with the findings.
The research method is well written. Experimental research method was used in the research.
Care was taken to write the tables used in the research in the form of APA6 standard.
Reliability values ​​in Table 3 should be given separately for each measurement tool. There is confusion there and it is not written according to 2.4. The variable names in Table 7 should also be written in accordance with 2.4.1. Variable names should not be written differently.
In Table 3, it is said "The average IOC score was 1.00". What is IOC here? Also 1 average for a test is very small.
" (Mean=31.61.43, SD=4.856)" was said but I did not see ever two dots for a decimal number.
" (Mean=53.29, SD=3.6662) " was said but after dot use 3 decimal places for SD.
The normal distribution of data is very important for this research. Any data collection greater than 30 does not guarantee normality. The skewness value and error rate can yield results. I suggest the authors check this out.
In Table 7, 4 zeros are not significant. It should be corrected. Actually sig. values ​​can be written as 0.001.
The results and recommendations section of the research are written in numbered form. It would be better if these sections were written in paragraphs.
Round 2
Reviewer 1 Report
The paper was fully revised, and my questions were completely answered.
So, I believe the article is ready for publication, except for the minor faults listed:
~ I believe figure 5 (page 9) do not add anything new to the text, it just shows an Excel facility probably everybody already knows.
~ Minor English typos, please revise these sentences
. page 2 "each teacher has access to if for use in classes"
. page 12 "before the formal test: Index of Congruence (IOC) and were determined from their responses and reliability values were determined by SPSS"
. page 12 "The experiment was used an experimental group taught"
. page 12 "who learnt with a traditional method , The WP scale test and"
. page 12 "and cognitive load for the both groups"
Reviewer 2 Report
The authors' amendment is generally consistent with my previous comments.
However, there is one more issue that may not be addressed as expected. Regarding the Shapiro-Wilk test. Yes, your result is suitable. However, I would like to suggest to the authors that they justify the suitability of Shapiro-Wilk for smaller sample sizes as opposed to Kolmogorov-Smirnov, which is significantly more appropriate for larger sample sizes.
Thank you.
Reviewer 3 Report
Most of the corrections are done. But there is one thing that should be corrected too. Last time I said:
"The normal distribution of data is very important for this research. Any data collection greater than 30 does not guarantee normality. The skewness value and error rate can yield results. I suggest the authors check this out."
But this time most of the Shapiro Wilks significant values are less than 0.05 and here parametric Anova tests were yielded in Table 4. This is a contradiction. If the data is not normally distributed, then other nonparametric tests should be done but not the Manova test. Also, for 55 average 25 standard deviation is too high. Authors can check it.
